

# Fungal diversity in shade-coffee plantations in Soconusco, Mexico

Eugenia Zarza[1,2], Alejandra López-Pastrana[3], Anne Damon[3], Karina Guillén-Navarro[1] and Luz Verónica García-Fajardo[1]

[1] Departamento de Ciencias de la Sustentabilidad, El Colegio de la Frontera Sur, Tapachula, Chiapas, Mexico
[2] Investigadora-CONACYT, Consejo Nacional de Ciencia y Tecnología, Ciudad de México, Mexico
[3] Departamento de Conservación de la Biodiversidad, El Colegio de la Frontera Sur, Tapachula, Chiapas, Mexico

## ABSTRACT

**Background**. As forested natural habitats disappear in the world, traditional, shade-coffee plantations offer an opportunity to conserve biodiversity and ecosystem services. Traditional coffee plantations maintain a diversity of tree species that provide shade for coffee bushes and, at the same time, are important repositories for plants and animals that inhabited the original cloud forest. However, there is still little information about shade-coffee plantation's fungal diversity despite their relevance for ecosystem functioning as decomposers, symbionts and pathogens. Specifically, it is unknown if and what mycorrhizae-forming fungi can be found on the branches and trunks of coffee bushes and trees, which hold a diversity of epiphytes. Here, we evaluate fungal communities on specific plant microsites on both coffee bushes and shade trees. We investigate the ecological roles played by this diversity, with a special focus on mycorrhizae-forming fungi that may enable the establishment and development of epiphytic plants.

**Methods**. We collected 48 bark samples from coffee bushes and shade trees (coffee; tree), from four plant microsites (upper and lower trunks, branches and twigs), in two shade-coffee plantations in the Soconusco region in southern Mexico, at different altitudes. We obtained ITS amplicon sequences that served to estimate alpha and beta diversity, to assign taxonomy and to infer the potential ecological role played by the detected taxa.

**Results**. The bark of shade trees and coffee bushes supported high fungal diversity (3,783 amplicon sequence variants). There were no strong associations between community species richness and collection site, plant type or microsite. However, we detected differences in beta diversity between collection sites. All trophic modes defined by FUNGuild database were represented in both plant types. However, when looking into guilds that involve mycorrhizae formation, the CLAM test suggests that coffee bushes are more likely to host taxa that may function as mycorrhizae.

**Discussion**. We detected high fungal diversity in shade-coffee plantations in Soconusco, Chiapas, possibly remnants of the original cloud forest ecosystem. Several mycorrhiza forming fungi guilds occur on the bark of coffee bushes and shade trees in this agroecosystem, with the potential of supporting epiphyte establishment and development. Thus, traditional coffee cultivation could be part of an integrated strategy for restoration and conservation of epiphytic populations. This is particularly relevant for conservation of threatened species of Orchidaceae that are highly dependent on mycorrhizae formation.

Corresponding authors
Anne Damon, adamon@ecosur.mx
Karina Guillén-Navarro,
kguillen@ecosur.mx

## INTRODUCTION

Biodiversity is severely threatened by human activities that have transformed natural habitat into agricultural and urban landscapes. Contrary to intensified monocultures, complex agroecosystems, such as shade-coffee plantations, offer a significant opportunity for the conservation of biodiversity (*Solís-Montero, Flores-Palacios & Cruz-Angón, 2005*; *Perfecto et al., 2007*) and ecological services like carbon sequestration, soil protection and conservation of native pollinators (*Toledo & Moguel, 2012*). Additionally, they are low maintenance and low impact agroecosystems, that do not require irreversible changes in their management (*Espejo Serna et al., 2005*; *García-González et al., 2011*) and that provide livelihoods for millions of people in more than 50 countries (*Wrigley, 1988*; *Vega, 2008*; *Amrouk, 2018*).

Traditional coffee plantations maintain a diversity of native tree species to shade the coffee bushes and are important repositories of the original biodiversity of the cloud forest ecosystem (*Moguel & Toledo, 1999*). Previous studies have shown that these agroecosystems harbor a high diversity of arthropods (*Méndez-Castro & Rao, 2014*; *Wauters, Fournier & Dekoninck, 2017*; *Ibarra-Isassi et al., 2021*), birds (*Mas & Dietsch, 2004*; *Tejeda-Cruz & Sutherland, 2004*; *Philpott et al., 2008*; *MacGregor-Fors et al., 2018*; *González et al., 2020*), mammals (*Gallina, Mandujano & González-Romero, 1996*; *Moguel & Toledo, 1999*; *Caudill, De Clerck & Husband, 2015*; *Etana et al., 2021*) and plants (*Cruz-Angón & Greenberg, 2005*; *González Zamora, Esperón Rodríguez & Barradas, 2016*; *Álvarez-Álvarez et al., 2021*).

On the other hand, there is still little information about fungal diversity in traditional coffee plantations despite their relevance for ecosystem functioning as decomposers, symbionts and pathogens (*Mueller, Bills & Foster, 2004*; *Nguyen et al., 2016*). Mycorrhiza forming fungi, essential for the establishment and development of some plants, have been detected in soil from shade-coffee plantations (*Rao et al., 2020*; *Jurburg, Shek & McGuire, 2020*; *Díaz-Ariza, Rivera & Sánchez, 2021*). Specifically, it is unknown if, and what mycorrhizae-forming fungi can be found on the branches and trunks of coffee bushes and shade trees, which hold a diversity of epiphytes, some of them endangered, as is the case of some orchids (*Moguel & Toledo, 1999*; *Solís-Montero, Flores-Palacios & Cruz-Angón, 2005*; *Espejo Serna et al., 2005*; *Mondragón, Santos-Moreno & Damon, 2009*; *Damon, 2017*).

Although we do not understand the mechanisms involved, it is clear that epiphytic orchids have evolved a complex and obligate dependence on certain species of fungi (*Selosse et al., 2022*). Various authors have demonstrated that the seeds of epiphytic orchids depend upon mycorrhizae forming endophytic fungi to enable the differentiation of the mother cells in the protocorm that develops from the embryo of the seed, thereby completing the process of germination, and that growth ceases and the plant eventually dies in the absence of a mycorrhizal symbiont (*Zettler, Delaney & Sunley, 1998*; *Markovina & McGee, 2000*; *Pereira et al., 2005*). After the initial germination and development stages, the association appears to become facultative and highly variable across species, seasons and distributions

on a local, but not on a geographical scale (*McCormick, Whigham & Canchani-Viruet, 2018*) and further research is required to establish patterns and tendencies.

These fungi can be observed as hyphal coils, or pelotons, within cells of the roots and are therefore termed endophytes, but furthermore they form an intimate mycorrhizal association with the orchid plant. We use the term mycorrhizae forming endophytic fungi, instead of the more usual term Orchid Mycorrhizal Fungi (OMF); first, to distinguish them from the networks of ectomycorrhizal fungal that associate with the external surfaces of plant roots, especially trees, and second, to avoid the idea that the fungi that associate with Orchidaceae form a small, well-defined group of fungal species.

Epiphytes germinate and complete their development on the bark of the branches, trunks and even the twigs of trees, bushes and lianas in forest ecosystems and we can assume that the necessary fungi are present on the bark surfaces and that the spatial distribution and niche preferences of these potentially mycorrhizal fungi will then influence the distribution of epiphytic orchids on those surfaces. Complex, variable and incompletely defined spatial distributions have been observed for endophytic mycorrhizal fungi associated with terrestrial orchids in soils and epiphytic orchids on bark surfaces. The OMF (equivalent to our mycorrhizae forming epiphytic fungi) evaluated by *Petrolli et al. (2021)* were spatially related to orchid roots, however, to the contrary, *Kartzinel, Trapnell & Shefferson (2013)* found that sowing the seeds of a rare epiphytic orchid close to or far from mature conspecific plants did not affect germination success. It should be noted that *Petrolli et al. (2021)* focused on young trees, and only two individuals, and as commented by *Kartzinel, Trapnell & Shefferson (2013)*, we know that the microbial community and environmental conditions found on older trees tend to be more favorable for the establishment of epiphytic orchids.

Here, we evaluate fungal communities on specific plant microsites on both coffee bushes and shade trees. We investigate the ecological roles played by this diversity, with a special focus on mycorrhizae-forming fungi that may enable the establishment and development of epiphytic plants.

## MATERIALS & METHODS

### Study area

The study area is in the southeast of Mexico, within a biodiversity hotspot (*Mittermeier et al., 2011*; *CONABIO, 2012*) including the Tacaná-Boquerón Biological Corridor, considered to be the second most important region for orchid species richness in Mexico (*Arriaga et al., 2000*; *Solano-Gómez et al., 2016*). Samples were collected from two localities, with mountainous, sloping terrain, separated by more than 50 km, in the Soconusco region in the state of Chiapas and chosen because of their differing characteristics (*e.g.*, altitude), in April 2017. "Los Hermanitos" (hereinafter referred to as H) is situated at 15°06′02.8″N and 92°19′14.9″W, at 537 m.a.s.l., towards the lowest altitude tolerated by *Coffea arabica*, whereas "Benito Juárez El Plan" (hereinafter referred to as B) is situated at 15°05′0.517″N and 92°08′14.806″W, and at a much higher elevation, 1,500 m.a.s.l. In each locality, four coffee bushes and two shade trees (hereinafter coffee, tree, respectively) were selected,

with four microsites each: twig, branch, upper trunk (near the origin of the first branch), and lower, or base of the trunk. In both sites, the overstory, including the trees sampled, were mature *Inga micheliana* (Fabaceae) and the original forest trees would have first been thinned out about 80 years ago and then replaced by *I. micheliana* within the last 20 or 30 years, although in most coffee plantations in the region some original forest tree species are retained. A $6 \times 7$ cm bark sample, 2–5 mm thick, was taken from each microsite ($n = 48$), using a previously disinfected scalpel. Each bark sample was individually placed in a labeled plastic bag and transported to the laboratory in an icebox with ice. We obtained verbal permission to collect samples from the land owners: Mr. Marco Polo Zamora Martínez, Predio Las Bugambilias, Finca Los Hermanitos, Tapachula, Chiapas; Mr. Bonifasio Morales Ortíz, field in Ejido Benito Juárez El Plan, Cacahoatán, Chiapas.

## Laboratory procedures

To conserve the microorganisms present on the bark surface, the samples were not surface sterilized. Genomic DNA was extracted from the surface of the 48 bark samples following a CTAB (Hexadecyltrimethylammonium bromide) based protocol (*Díaz Cárdenas et al., 2008*). DNA quality was checked on a 1% agarose gel, stained with SYBR® Green and visualized under UV light. Using the ITS3F (GCATCGATGAAGAACGCAGC ) and 4R (TCCTCCGCTTATTGATATG) primer pair (*White et al., 1990*), an ITS2 amplicon library was built and run on an Illumina MiSeq instrument at Macrogen Inc. in Seoul, South Korea to produce 300 bp paired-end sequences.

## Bioinformatics and statistical analyses

Sequence quality was assessed with FastQC v0.11.8 (*Andrews, 2018*) and summarized with MultiQC (*Ewels et al., 2016*). Adapters and primer fragments were removed with CUTADAPT version 1.18 (*Martin, 2011*). The resulting trimmed sequences were imported into and analyzed with the package QIIME2 version q2cli 2019.1.0 (*Bolyen et al., 2019*). The ITSxpress plugin was used to remove the conserved regions flanking ITS2 to improve accuracy in taxonomic assignment (*Rivers et al., 2018*) and considering models for all taxa. Although the amplicon library was designed to amplify fungal markers, this 'all taxa' approach was applied to give a glimpse of the eukaryotic diversity associated with the selected microsites. The DADA2 (*Callahan et al., 2016*) plugin was used to de-noise sequences, correct errors in marginal sequences, join paired-end reads and remove chimeric and singleton sequences. We used default values for parameters, except for removing reads with more than six errors (max-ee = 6). The DADA2 plugin produced a feature table and a file with representative sequences (*i.e.*, amplicon sequence variants -ASVs). Results and statistics were visually assessed with QIIME2-view (https://view.qiime2.org/). All ASVs were classified using the feature-classifier plugin and using the UNITE 8.0 database (*Nilsson et al., 2019*) for QIIME version 18.11.2018 with trimmed reference sequences for all eukaryotes (https://dx.doi.org/10.15156/BIO/786335). The QIIME classifier was trained with this database and the 'fit-classifier-naive-bayes' method. Representative sequences were then classified using the 'sklearn' method. Heatmaps were generated with R (*R Core Team, 2017*) to better visualize diversity at class level.

Alpha (Shannon Index) and beta diversity (Bray–Curtis distance) metrics were calculated with the q2-diversity plugin, which rarefies according to a user specified depth, aiming to choose a value that is as high as possible while excluding as few samples as possible. Additionally, tests for associations between alpha diversity and collection site, plant type, microsite type (Group 1–G1 from now onwards), and plant type-microsite combinations (Group 2–G2 from now onwards) were carried out. Similar associations were investigated but using beta diversity parameters.

Modifications to this pipeline were implemented to analyze diversity patterns in Fungi only. To that end, ITSxpress was run again specifying Fungi as the focus taxa. Taxonomic classification was carried out with the previously trained classifier. The resulting table was used to filter the feature table and include only sequences assigned to the Fungi kingdom. Alpha (Shannon Index) and beta diversity (Bray–Curtis distance) metrics were calculated using this filtered table as input. For these calculations we used the parameter 'sampling-depth' implemented in QIIME2 to randomly subsample the counts in each sample, without replacement, so that each sample in the resulting table has even sampling. Following QIIME2 guidelines and after checking the filtered feature table, we chose a sampling depth value that was as high as possible while aiming to lose as few samples as possible.

The feature table containing only Fungi sequences was imported into R to perform further diversity analyses and create visualizations. The library ampvis2 v2.7.27 (*Andersen et al., 2018*) and amplicon v1.14.2 (*Liu et al., 2021*) were used to filter samples with a minimum of 10,000 reads and rarefy at two different depths (20,000 and 12,408 reads), calculate alpha diversity and create box plots. To test if there were significant differences between alpha diversity among collection sites, plant type, microsite type (Group 1), and plant type-microsite combinations (Group 2) we performed a Tukey range test. Additionally, to visualize differences in beta diversity among groups (same as above), we performed a non-metric multidimensional scaling (NMDS) analysis based on the Bray–Curtis distance and tested for differences in fungal composition among groups with the PERMANOVA method, applying the adonis function implemented in vegan (*Oksanen et al., 2018*) and included in the package amplicon.

To obtain a list of species that are associated with particular microsites (or combinations of those), we performed an indicator species analysis using the R package indicspecies v1.7.12 (*De Cáceres & Legendre, 2009*) with the function multipatt executing 999 permutations.

Sequences classified with QIIME2 were submitted to the FUNGuild v1.0 database (*Nguyen et al., 2016*) to assign functional annotation. We created boxplots with R (*R Core Team, 2017*) to visualize the trophic mode distribution on the different microsites. To investigate a possible association between fungal taxa with known mycorrhizal potential and coffee bushes or shade trees, we applied the CLAM test (*Chazdon et al., 2011*) as implemented in the R v.3.4.1 program vegan v2.5-3 (*Oksanen et al., 2018*), to perform a classification of habitat specialists/generalists, and those too rare to classify, in two distinct habitat types (*i.e.*, coffee and trees), minimizing bias due to differences in sampling intensities. We classified all ASVs identified as belonging to the Fungi kingdom, Fungi

genera and mycorrhizal guilds. We used a coverage limit of 10, a 'supermajority' (2/3) specialization threshold, and a significance alpha value of 0.0025 to correct for multiple comparisons.

# RESULTS

DNA was successfully purified from all 48 samples. The number of sequences obtained, number of ASVs, microsite type and plant type are shown in File S1. Between 142,544–256,570 sequences were obtained for each sample. After ITSxpress trimming for all taxa, filtering, and DADA2 denoising, between 9,357–140,012 sequences were retained. When selecting only Fungi with ITSxpress, followed by filtering and DADA2 denoising, between 2,884–128,158 sequences were retained.

Diversity metrics obtained with QIIME2 for the 'All taxa' dataset were calculated at a sampling depth of 77,000 reads. At this depth, the alpha-rarefaction curves plateaued, indicating that sampling was representative for all plant-microsite treatments, except in 'Tree Twigs' (File S1, Fig. S1). Group significance tests suggested that there was no strong association between community richness (*i.e.*, Shannon Index) and collection site (B and H), plant type (coffee; tree), G1 (microsite) and G2 (plant type-microsite) combinations. To detect if samples within a group were more similar to each other than to samples from other groups, PERMANOVA pairwise tests were carried out. To determine which specific pairs differed from one another we used the Bray–Curtis distance obtained with QIIME2. Significant differences were obtained when comparing collection site (B *vs.* H, $p = 0.004$) and plant type (coffee; tree, $p = 0.001$). No significant differences were obtained within G1 nor within G2. Visualizations for alpha and beta diversity associations are presented in File S1, Fig. S3.

Some of the sequences that passed the filter for only Fungi sequences applied by ITSxpress (*i.e.*, 4414 ASVs), were not actually classified as belonging to the Fungi kingdom, according to the QIIME2 classifier (Files S2 and S3). They included Alveolata (53), Chromista (14), Protista (254), Rhizaria (5), Viridiplantae (5), and unassigned sequences (300). Thus, another filter was applied in QIIME2 to consider only ASVs tagged as Fungi resulting in 3783 ASVs. Only 75% of these Fungi ASVs were assigned to phylum, 29% to genus, and 16% to species level. The most widespread fungi, classified with the BLAST tool implemented in QIIME2-view, (*i.e.*, detected in 25%–45% of samples) were *Cladosporium tenuissimum*, *Lasiodiplodia*, *Fusarium* and *Pestalotiopsis*, which are considered as common pathogens. On the other hand, the species indicator analysis detected 11 ASVs associated with only five microsites, or their combinations on the same plant type. All except one (*i.e.*, *Bulleribasidium*), belonged to Ascomycota, and included several pathogens, endophytic and lichen forming taxa (Table 1). Three occurred in coffee bushes and the rest in shade trees. A further 29 ASVs were associated with 11 microsites or their combinations, however they could not be taxonomically assigned beyond family level (File S2).

Regarding this fungal diversity, alpha-rarefaction curves plateaued when considering a sequencing depth of 20,000, indicating that sampling was representative for all plant-microsite combinations (Fig. S2). Comparisons performed with the Tukey test did not

**Table 1** **Fungal taxa associated with bark microsites in coffee bushes and shade trees according to an indicator species analysis.** Only taxa that were classified to genus and species are shown.

| Microsite | Taxon | Description |
|---|---|---|
| CTw | A: Glomerellales, *Colletotrichum gigasporum* | Genus globally distributed, on various plants as epiphyte, saprobe, endophyte and pathogen. It has been reported on coffee (*Douanla-Meli & Unger, 2017*). |
| | B: Tremellales, *Bulleribasidium* | 11 species, some parasitize other fungi (e.g., *Cladosporium sp.*) (*Sampaio et al., 2002*). |
| CBr, CTh, CTl | A: Hypocreales, *Tolypocladium* | Parasites of truffle-like fungi, soil saprotrophs, plant endophytes, pathogens of insects, nematodes, rotifers (*Borel et al., 1976*; *Yu et al., 2021*). |
| TTw | A: Chaetothyriales, *Strelitziana africana* | Isolated from leaves of Strelitzia in South Africa (*Arzanlou & Crous, 2006*). |
| | A: Capnodiales, *Cladosporium* | Plant pathogens, fungi parasites, soil inhabitants. Spores are wind dispersed and abundant (*Parbery, 1969*). |
| | A: Ostropales, *Absconditella rubra* | Genus of lichenized fungi (*Czarnota & Kukwa, 2008*). |
| | A: Capnodiales, *Pseudocercospora norchiensis* | Leaf pathogens on many economically and ecologically important plant species, including *Eucalyptus* and neotropical species (*Pérez et al., 2013*). |
| | A: Pleosporales, *Periconia* | Plant and occasional human pathogens, saprobes. Genus reported in cloud forest of Veracruz, Mexico (*Arias Mota & Heredia Abarca, 2020*). |
| TBr | A: Xylariales, *Pestalotiopsis* | Plant pathogens; they have been reported on coffee plants (*Song et al., 2013*). |
| | A: Xylariales, *Biscogniauxia* | Some species are tree pathogens (*Nugent et al., 2005*). |
| TBr, TTw | A: Xylariales, *Pseudopestalotiopsis simitheae* | Isolated from *Pandanus*; pathogenes, considered a threat to tropical hosts; reported in a wide range of hosts across the world (*Nozawa et al., 2017*; *Gualberto et al., 2021*). |

**Notes.**
A, Ascomycota; B, Basidiomycota; CTw, coffe twig; CTh, coffee trunk high; CTl, coffee trunk low; CBr, coffee branch; TTw, tree twig; TBr, tree branch.

detect any significant difference in alpha diversity (Shannon and Simpson indices), among microsites when rarefying at 12,408 (Fig. S3) and 20,000 (Fig. 1). Similarly, there were no strong associations between community species richness, and any of: collection site, plant type, G1 or G2, as calculated with QIIME2 with the Shannon index and the Kruskal–Wallis test.

On the other hand, significant differences in species composition were obtained when comparing collection sites (B *vs.* H, $p = 0.001$) with the Bray–Curtis distance and the PERMANOVA method implemented in QIIME2. No significant differences were obtained between plant types, within G1, or within G2 (File S1, Figs. S4–S6). These results were confirmed by the NMDS and adonis tests implemented in R (Fig. 2). There was a clear separation between the samples collected in the two localities (adonis $R = 0.036$, $p = 0.001$). Significant results were also obtained when comparing groups defined by plant type (coffee; trees, adonis $R = 0.026$, $p = 0.002$), however there was an overlap between these groups when observing the NMDS graph (Fig. 2B). When comparing branch, twig, trunk high and trunk low, there were only significant differences when 'twig' was part of the comparison (twig *vs.* trunk low, adonis $R = 0.055$, $p = 0.001$; twig *vs.* trunk high, adonis $R = 0.052$, $p = 0.005$; twig *vs.* branch adonis $R = 0.048$, $p = 0.08$). When comparing

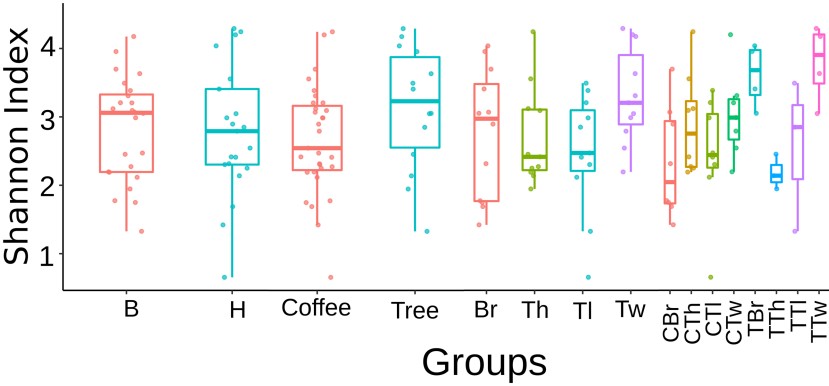

**Figure 1** Boxplots showing Shannon Index for Fungi alpha diversity detected on bark samples, according to collection site, plant type, sampled plant part and microsite in two shade-coffee plantations. B, Benito Juárez El Plan; H, Hermanitos; CTw, coffee twig; CTh, coffee trunk high; CTl, coffee trunk low; CBr, coffee branch; TTw, tree twig; TTh, tree trunk high; TTl, tree trunk low; TBr, tree branch.

microsites, we detected significant differences in beta diversity when comparing the twigs and trunks of coffee bushes (coffee twig *vs.* coffee trunk low, adonis $R = 0.08, p = 0.001$; coffee twig *vs.* coffee trunk high, adonis $R = 0.08, p = 0.012$), and when comparing tree twig *vs.* tree trunk low (adonis $R = 0.16, p = 0.027$). Similarly, beta diversity between tree twig and other sites on coffee bushes was significantly different (tree twig *vs.* coffee trunk high, adonis $R = 0.105, p = 0.044$; tree twig *vs.* coffee branch, adonis $R = 0.105, p = 0.003$; coffee trunk low vs tree twig, adonis $R = 0.106, p = 0.005$). We also detected differences in beta diversity between branches of both plant types (tree branch *vs.* coffee branch, adonis $R = 0.101, p = 0.019$). Beta diversity of samples collected from coffee trunk low and high was significantly different from beta diversity of tree branches (adonis $R = 0.101, p = 0.013$; adonis $R = 0.103, p = 0.05$). Beta diversity was also significantly different between coffee trunk high and tree trunk low (adonis $R = 0.1, p = 0.035$).

A heatmap at class level was created to evaluate possible trends in taxon frequency, according to sample, plant type or microsite (Fig. 3). Agaricomycetes (Basidiomycota) was the most relatively abundant class (52.8% of total reads), particularly in a cluster containing mostly coffee bush samples (lower cluster in dendrogram). These fungi comprise a wide spectrum of ecological functions. The second most abundant (19.47%) were Dothideomycetes (Ascomycota), mostly endophytic or saprobes, with higher abundance in the upper cluster that includes both coffee bush and shade tree bark samples. Other Ascomycota classes that are relatively abundant in samples in the upper cluster and have several ecological functions were Sordariomycetes (8.25%), Eurotiomycetes (5.11%), Lecanoromycetes (4.99 %), Leotiomycetes (3.38%), Orbiliomycetes (3.33 %), together with Tremellomycetes (1.48%, Basidiomycota).

ASVs considered as Fungi, were assigned to nine trophic modes using FUNGuild. All trophic modes were sampled in both plant types (Fig. 4), except pathogen-saprotroph-symbiont (only on coffee twigs and trunk) and saprotroph-pathotroph-symbiotroph (on trunk lower area). However, these two annotations might be redundant in the FUNGuild

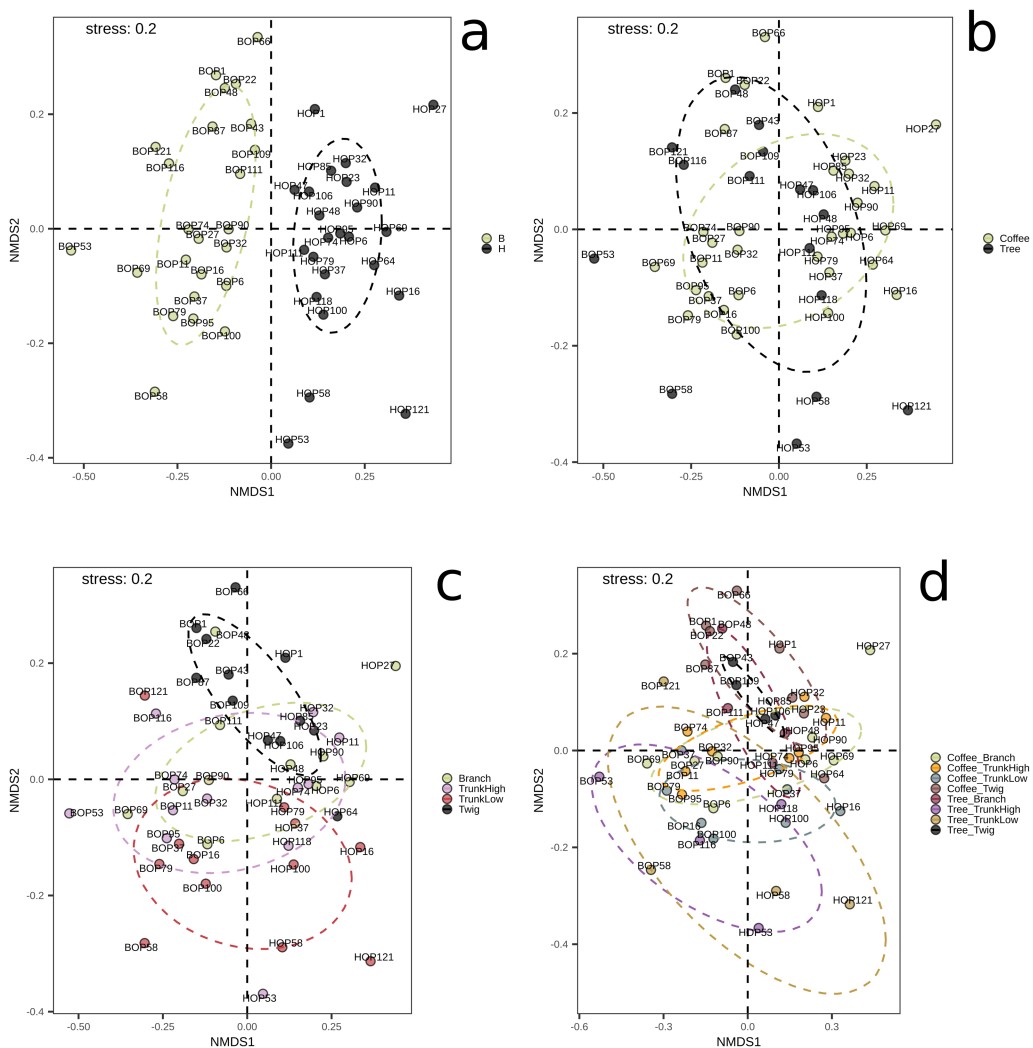

**Figure 2  NMDS plots showing groupings of bark samples according to (A) collection site; (B) plant type; (C) sampled plant part; and (D) microsite.** Notice the clear separation associated with the collection site. B, Benito Juárez El Plan; H, Hermanitos.

database and actually belong to the pathotroph-saprotroph-symbiont classification. Symbiotrophs show higher relative abundance in coffee bushes than in shade trees –except for tree twigs. When looking into guilds that involve mycorrhizae formation, some occur only on coffee bushes or have a higher frequency than in samples from shade trees (Fig. 5). This may suggest that, although both types of plants harbor similar diversity, coffee bushes are more likely to host taxa that may function as mycorrhizae. All orchid mycorrhiza sequences were detected on 'coffee trunk low', with a relative sequence abundance of 0.0036. Table 2 shows the total occurrence of sequences classified as mycorrhizal fungi, stating the lowest common taxonomy assigned and ecological description.

We applied the CLAM test to only include ASVs tagged as belonging to the Fungi Kingdom (3,783) resulting in 1,648 ASVs classified as coffee bush specialists and 1,099 as

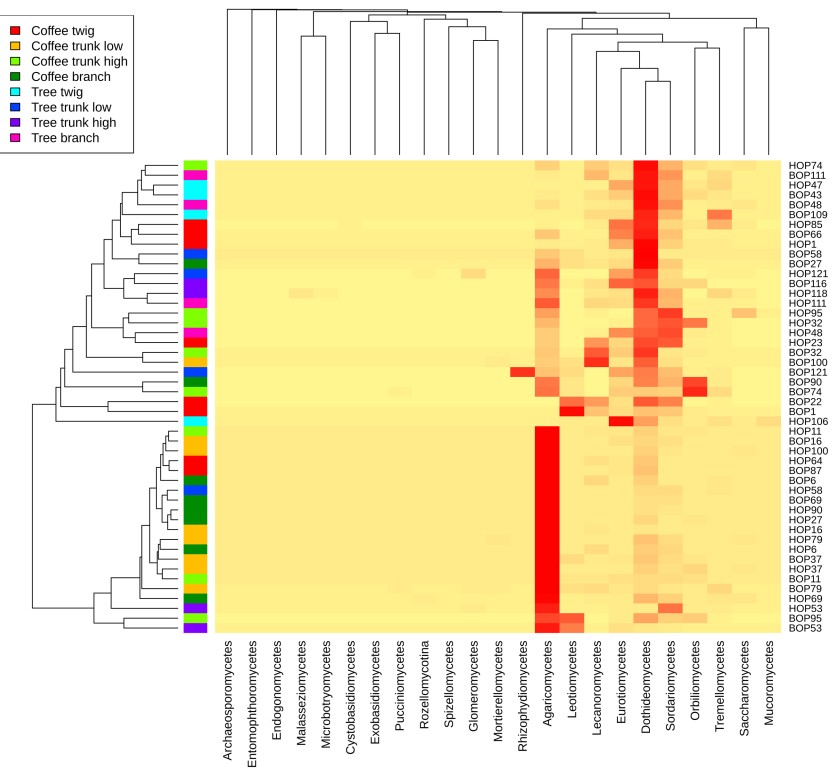

**Figure 3** Heatmap showing the frequency of sequences assigned to Fungi Classes in samples taken from coffee bushes and shade trees in two shade-coffee plantations in Soconusco, Chiapas, Mexico.

shade tree specialists, representing 43.6% and 29.1%, respectively. Only 3.8% ASVs were classified as generalists and 23.6% as too rare to be classified. On the other hand, 111 of the 302 identified genera were classified as coffee bush specialists and 85 as shade tree specialists, with 58 being generalist and 49 too rare to be classified. We were also interested in investigating if any of the mycorrhizal guilds could be classified as coffee bush or shade tree specialists. The CLAM test suggests that 50% (6 guilds) of the mycorrhizal fungi can be classified as coffee bush specialists and 25% (3 guilds) as shade tree specialists, whereas two guilds are generalists and one is too rare to be classified. Table 3 shows the mycorrhizal guilds and their classification.

The data that support the findings of this study are openly available in GenBank at https://www.ncbi.nlm.nih.gov/genbank/, under the Bioproject PRJNA610266. Commands and scripts used to run bioinformatics analyses are available from https://github.com/zarzamora23/Fungi-ITS-analyses.git.

## DISCUSSION

An understanding of the microbiome of agroecosystems is essential to guide management strategies and ensure sustainability (*Toju et al., 2018*). Some epiphytic plants (*e.g.,* orchids) are primarily dependent upon mycorrhiza-forming endophytic fungi and later dependent upon other organisms, such as specialist pollinators, to establish stable, persistent

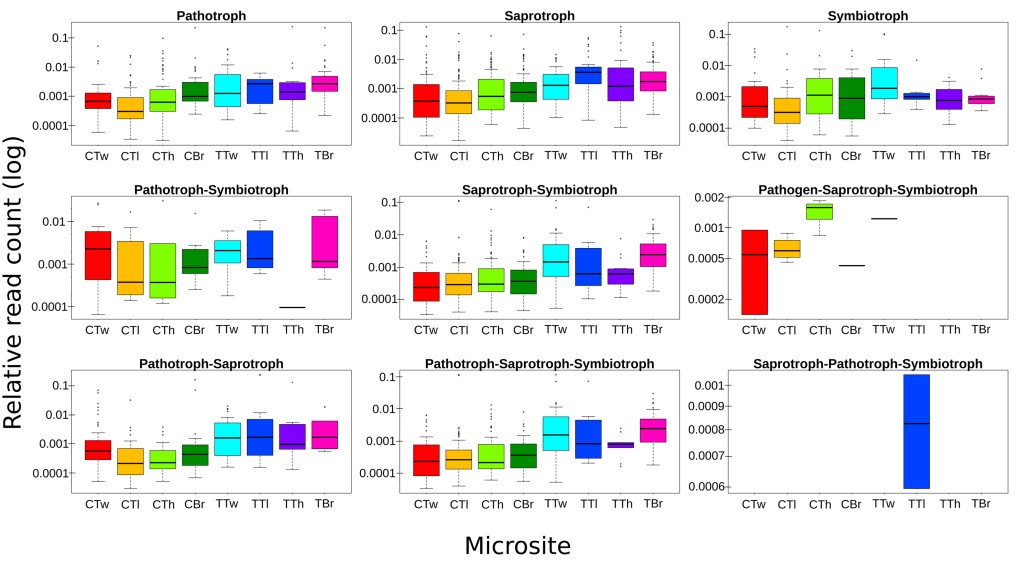

**Figure 4   Boxplots showing trophic mode frequency per microsite according to FUNGuild analysis.**
Bark samples were taken from coffee bushes and shade trees in two shade-coffee plantations in Soconusco, Mexico. Relative read abundance given in log scale.

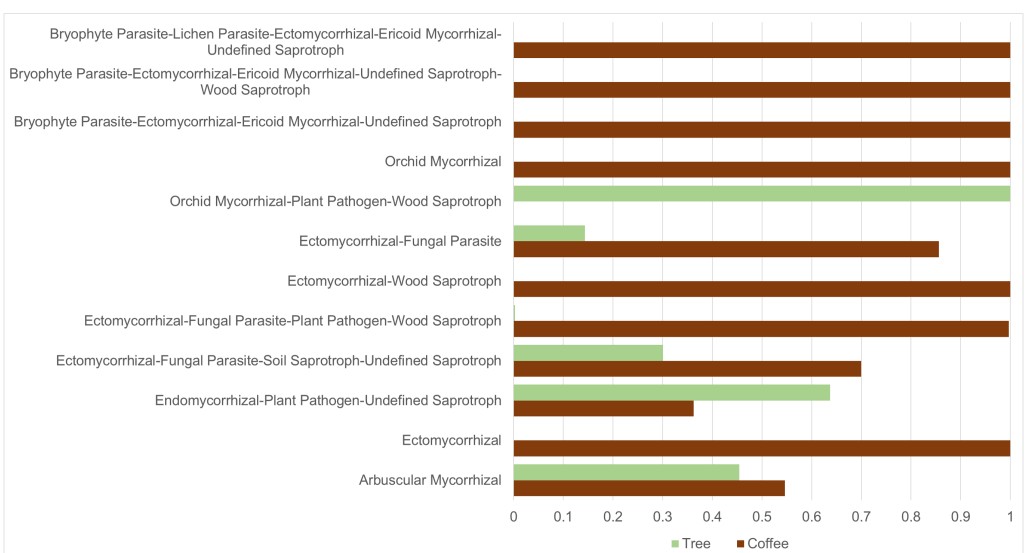

**Figure 5   Mycorrhizal guild frequency per plant type, detected in bark samples obtained from four microsites on coffee bushes and shade trees in two shade-coffee plantations in Soconusco, Chiapas, Mexico.**

populations. Traditional coffee plantations, which host a variety of native species of trees, offer a unique opportunity for the conservation of epiphytic plants and other elements of the original native flora and fauna. However, recently, in the region of Soconusco, management practices designed to increase production (*e.g.*, monoculture shade, heavy

**Table 2** Total occurrence of sequences classified as mycorrhizal fungi by FUNGuild in bark samples from four microsites on coffee bushes and shade trees in two shade-coffee plantations in Soconusco, Mexico.

| Sequences | Guild | Lowest taxonomy | Microsite |
|---|---|---|---|
| 8,555 | Ectomycorrhizal-Fungal Parasite-Plant Pathogen-Wood Saprotroph | Helotiaceae | CTw,CTl,CTh,CBr,TTl |
| 6,667 | Endomycorrhizal-Plant Pathogen-Undefined Saprotroph | Ceratobasidiaceae | CTw,CTl,CTh,CBr,TTl,TTh,TBr |
| 4,287 | Ectomycorrhizal-Fungal Parasite | *Delicatula integrella* | CTl,TTl |
| 1,649 | Arbuscular Mycorrhizal | Diversisporales (*Acaulospora lacunosa*), Glomerales (*Rhizophagus*), Archaeosporales, Gigasporales | CTl,CTh,TTl,TTh |
| 637 | Orchid Mycorrhizal | *Serendipita* | CTl |
| 429 | Ectomycorrhizal-Fungal Parasite-Soil Saprotroph-Undefined Saprotroph | *Entoloma* | CTw,CTl,CTh,TTh,TBr |
| 120 | Bryophyte Parasite-Ectomycorrhizal-Ericoid Mycorrhizal-Undefined Saprotroph | *Rhizoscyphus monotropae* | CTl |
| 117 | Bryophyte Parasite-Lichen Parasite-Ectomycorrhizal-Ericoid Mycorrhizal-Undefined Saprotroph | *Pezizella* | CTl,CTh,CBr |
| 106 | Ectomycorrhizal-Wood Saprotroph | *Tropicoporus linteus* | CTl,CTh |
| 97 | Orchid Mycorrhizal-Plant Pathogen-Wood Saprotroph | *Rhizoctonia fusispora* | TTl |
| 93 | Ectomycorrhizal | Sebacinales, *Endogone* | CTl, CTh |
| 10 | Bryophyte Parasite-Ectomycorrhizal-Ericoid Mycorrhizal-Undefined Saprotroph-Wood Saprotroph | *Pezoloma ericae* | CTh |

Notes.
CTw, coffe twig; CTh, coffee trunk high; CTl, coffee trunk low; CBr, coffee branch; TTw, tree twig; TTh, tree trunk high; TTl, tree trunk low; TBr, tree branch.

pruning or elimination of shade trees, use of new, dwarf coffee varieties, removal of moss and epiphytes, and the use of agrochemicals) have been shown to endanger the local flora and fauna that had adapted to the original traditional, shaded coffee agroecosystem (*Perfecto et al., 1996*; *Moguel & Toledo, 1999*) during the one and a half centuries since the introduction of coffee into the Soconusco region. Here, we investigated fungal diversity on the coffee bushes themselves and on the trees providing shade in two traditional coffee plantations at different altitudes, to detect taxa with the potential to form mycorrhizal interactions with epiphytic plants.

Fungal diversity in general was high in both coffee bushes and shade trees in shade-coffee plantations in the Soconusco region. We identified 3,783 ASVs classified as Fungi, representing more OTUs than those detected in other bark studies carried out in temperate forests (*e.g.*, 2,207 by *Hagge et al., 2019*; 1,945 by *Pellitier, Zak & Salley, 2019*). However, the data are not entirely comparable as, unlike our study, they were interested in fungal colonization within the bark, and the external surface was sterilized prior to sampling. On the other hand, another study focusing on orchid mycorrhizal fungi in orchid roots and non-sterilized bark samples (trunk, fork and branch) of tropical, urban trees (*Izuddin et al., 2019*) detected 26 orchid mycorrhizal fungi, belonging to Ceratobasidiaceae, Serendipitaceae and Tulasnellaceae. The fork microsite had the highest abundance, whereas the branches had the highest diversity and the trunks the lowest. Ceratobasidiaceae were

**Table 3** CLAM classification of mycorrhiza forming fungi guilds as specialist, generalist or rare, detected in bark from coffee bushes and shade trees in two shade-coffee plantations in Soconusco, Mexico.

| Guild type | Coffee bushes (total sequences) | Shade trees (total sequences) | Classes |
|---|---|---|---|
| Arbuscular mycorrhizal | 900 | 749 | Specialist: tree |
| Ectomycorrhizal | 93 | 0 | Specialist: coffee |
| Endomycorrhizal, plant pathogen, undefined saprotroph | 2,418 | 4,249 | Specialist: tree |
| Ectomycorrhizal, fungal parasite, soil saprotroph, undefined saprotroph | 300 | 129 | Generalist |
| Ectomycorrhizal, fungal parasite, plant pathogen, wood saprotroph | 8,530 | 25 | Specialist: coffee |
| Ectomycorrhizal, wood saprotroph | 106 | 0 | Specialist: coffee |
| Ectomycorrhizal, fungal parasite | 3,671 | 616 | Generalist |
| Orchid mycorrhizal, plant pathogen, wood, saprotroph | 0 | 97 | Specialist: tree |
| Orchid mycorrhizal | 637 | 0 | Specialist: coffee |
| Bryophyte parasite, ectomycorrhizal, ericoid, mycorrhizal, undefined saprotroph | 120 | 0 | Specialist: coffee |
| Bryophyte parasite, ectomycorrhizal, ericoid, mycorrhizal, undefined saprotroph, wood saprotroph | 10 | 0 | Too rare |
| Bryophyte parasite, lichen parasite, ectomycorrhizal, ericoid mycorrhizal, undefined saprotroph | 117 | 0 | Specialist: coffee |

associated with the roots of three orchid species, Serendipitaceae with six and Tulasnellaceae with two orchid species. They also found non-*Rhizoctonia* fungi shared by bark and orchid roots and these could also prove to have mycorrhizal function (Ascomycetes–*Fusarium, Lachnum, Curvularia*; Basidiomycetes–*Mycena, Marasmius*).

Agaricomycetes and Dothideomycetes were the most relatively abundant classes on the collected bark samples. The latter was also highly abundant in a study analyzing the soil fungal community in coffee plantations and a nearby forest in El Salvador (*Rao et al., 2020*). Classes Sordariomycetes, Agaricomycetes, Lecanomycetes, Euromycetes were also highly abundant, coinciding with those detected in high abundance in our study (*Rao et al., 2020*). This reflects some similarities between fungal communities in coffee plantations in southern Mexico and coffee plantations and forests in El Salvador.

We detected several taxa forming endo, ecto and arbuscular mycorrhizal fungi belonging to 13 families (Table 2), suggesting that shade tree and coffee bush bark in the shade-coffee plantations could serve as reservoir of fungi needed for the establishment and development of epiphytic plants. We did not detect significant differences in alpha diversity among sites, plant type or microsite. However, there are significant differences in beta diversity among sites, plant type and some microsites. Differences in beta diversity, and thus in community composition, between sites could be associated with differences in altitude (*Gómez-Hernández et al., 2012*; *Ogwu et al., 2019*; *Arias Mota & Heredia Abarca, 2020*). It is possible that some species have more affinity for higher altitudes (537 *vs.* 1,500 m. a. s. l.), but sampling at intermediate sites would help to understand the effect of altitude and other environmental variables on fungal diversity and community composition. Distance from conserved or perturbed areas might be an important factor influencing

community composition as this may affect the re-colonization potential of the sites (*López-Quintero et al., 2012*; *Hazard et al., 2013*). Benito Juárez el Plan is situated 200 m away from areas still covered with original vegetation, whereas Los Hermanitos is several kilometers away. On the other hand, it is unlikely that time since land use changed has influenced beta diversity as both sites were transformed into shade-coffee plantations around the same time, more than 50 years ago.

The NMDS plot does not show a clear distinction between samples collected on coffee bushes and shade trees, but species composition was significantly differently according to the PERMANOVA and adonis tests. Fungi community composition in the microsite 'twig' showed significant differences from other microsites more often than other microsites, holding the highest number of exclusive taxa as demonstrated with the indicator species test. A possible explanation for differences in beta diversity in microsites and presence of indicator species might be a result of differences in bark texture of the coffee bushes and shade trees, as well as differences in water retention, pH or chemical composition that can be more favorable for some species of fungi than others (*Pecoraro et al., 2021*).

Interestingly, some of the 'indicator species' are considered as pathogens; this kind of function receives particular attention due to its economic importance and might be the reason why they could be identified. However, there were other 26 ASVs classified as indicator species in our analysis that could not be taxonomically assigned (File S2). In addition to the ecological roles that the different fungi species are playing in the coffee plantations, it is worth mentioning that some of them—or their relatives—have biotechnological potential (*e.g.*, *Tolypocladium*; *Borel, Kis & Beveridge, 1995*) which adds another ecosystem service provided by shade-coffee plantations in Soconusco, and other regions of the world, in areas previously covered by cloud forest. A biogeographic study of fungi in the Neotropical Cloud Forest (NTCF), resulted in a species list and distribution records comprising 2,962 species (*Del Olmo-Ruiz et al., 2017*). We detected 40 of those species in the shade-coffee plantations in Soconusco, which are probably a remnant of the diversity held by the original cloud forest. However, this is likely an underestimation as most of the ASVs in our study could not be assigned to species level. On the other hand, we detected 19 ASVs classified as Glomeromycota that comprise arbuscular mycorrhizae. Interestingly, *Del Olmo-Ruiz et al. (2017)*, defined an area of endemicity of cloud forest in the Chiapas Highlands based on the distribution of six species of Glomeromycota. The authors hypothesize that these arbuscular mycorrhizae fungi and their association with epiphytic bromeliads are relevant for community assembly in the cloud forest. *Del Olmo-Ruiz et al. (2017)* found that Ascomycota and Basidiomycota were the most abundant phyla in the cloud forest in the Mesoamerica region. These Fungi phyla were the most relatively abundant phyla in our study and in the Fungi community described by *Rao et al. (2020)* in soil samples from a shade-coffee plantation in El Salvador.

All trophic modes considered in the FUNGuild database are present in the collection sites and on microsites. As we did not sterilize the bark samples, the fungal community reflects the available organisms in the collection site and surrounding areas (*Ovaskainen et al., 2020*). Symbiotrophs have higher abundances on coffee bushes microsites than on shade trees. The presence of fungi with this trophic mode on bark is essential for

the establishment of epiphytic plants (*McCormick, Whigham & Canchani-Viruet, 2018*). Although, establishment also depends on the entire fungal community, for example competitors, other symbionts enabling plant growth, and the availability of nutrients produced by other fungi, *etc.* (*Pecoraro et al., 2018*).

Thus, shade-coffee plantations still hold some of the fungal diversity of the cloud forest, allowing the occurrence of important ecological processes such as mycorrhizal formation and epiphyte development. The type and number of mycorrhizal fungi detected depends greatly on taxonomic assignment, and this is limited by the available databases. In turn, this affects guild assignment and classification as specialists or generalists. Thus, it is likely that the diversity of mycorrhiza and other fungi we detected is an underestimation, which should serve to encourage further study of the ecology, taxonomy and molecular identification of these taxa.

## CONCLUSIONS

We detected high fungal diversity in shade-coffee plantations in Soconusco, Chiapas, possibly including remnants of the original cloud forest ecosystem. Several mycorrhiza forming fungi guilds occur on the bark of coffee bushes and shade trees in this agroecosystem, with the potential of supporting epiphyte establishment and development. We are working towards developing an integrated strategy for a return to traditional coffee cultivation (*Perfecto & Armbrecht, 2002*; *Harvey et al., 2008*), in protected areas in the region of Soconusco, with particular emphasis on the restoration and conservation of epiphytic plants (*e.g.*, orchids) within this agroecosystem (*Hietz, 2005*; *Toledo & Moguel, 2012*; *Toledo-Aceves et al., 2013*). It is paramount to restore microorganisms that facilitate and promote ecosystem health in general and the various stages of plant development.

## ACKNOWLEDGEMENTS

The authors thank Nelson Pérez Miguel and Fabiola Hernández Ramírez for their help during sample collection, and Ma. de los Ángeles Palomeque Rodas for labwork.

### Funding

Alejandra López-Pastrana was supported by a postgraduate scholarship from Consejo Nacional de Ciencia y Tecnología (CONACYT, No. 595182). The funders had no role in study design, data collection and analysis, decision to publish, or preparation of the manuscript.

### Grant Disclosures

The following grant information was disclosed by the authors:
Consejo Nacional de Ciencia y Tecnología: 595182.

## Competing Interests

The authors declare there are no competing interests.

## Author Contributions

- Eugenia Zarza analyzed the data, prepared figures and/or tables, authored or reviewed drafts of the article, and approved the final draft.
- Alejandra López-Pastrana conceived and designed the experiments, performed the experiments, analyzed the data, authored or reviewed drafts of the article, and approved the final draft.
- Anne Damon conceived and designed the experiments, authored or reviewed drafts of the article, and approved the final draft.
- Karina Guillén-Navarro conceived and designed the experiments, authored or reviewed drafts of the article, and approved the final draft.
- Luz Verónica García-Fajardo performed the experiments, analyzed the data, authored or reviewed drafts of the article, and approved the final draft.

## Field Study Permissions

The following information was supplied relating to field study approvals (*i.e.*, approving body and any reference numbers):

We obtained verbal permission to collect samples from the land owners: Mr. Marco Polo Zamora Martínez, Predio Las Bugambilias, Finca Los Hermanitos, Tapachula, Chiapas; Mr. Bonifasio Morales Ortíz, field in Ejido Benito Juárez El Plan, Cacahoatán, Chiapas.

## DNA Deposition

The following information was supplied regarding the deposition of DNA sequences:

The data are available in GenBank: PRJNA610266.

## Data Availability

Commands and scripts used to run bioinformatics analyses are available at GitHub: https://github.com/zarzamora23/Fungi-ITS-analyses.git.

## Supplemental Information

Supplemental information for this article can be found online at http://dx.doi.org/10.7717/peerj.13610#supplemental-information.

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
