# Peer review of "Fungal diversity in shade-coffee plantations in Soconusco, Mexico"

_PeerJ, doi:10.7717/peerj.13610_

## Round 0.1 · original submission · Major Revisions

We are sorry for the delay, but the pandemic meant that two original reviewers were not able to submit their comments. Please consider every issue raised by Reviewer 1, I totally coincide with comments, in particular, more important literature on previous research on mycorrhizae on barks should be included. Also, justification of the scarce sampling should be considered as well as additional statistical analyses and explain them carefully in methods and aligned with the proposed questions. Results of these analyses should be included as well as discussed. Discussion should be improved taking into account the new literature added.

Reviewer 1 ·

Basic reporting

Fungal diversity in coffee plantations was studied in a variety of different ‘habitat’ types on the coffee trees and bushes. This is a novel study and provides interesting information on understudied habitat types for an economically and culturally important plant species. The dataset is relatively rich, but additional analyses and statistical reporting are required for publication.

Introduction
Overall the introduction is sufficient. However, additional detail on the rationale for finding orchid mycorrhiza on bark is necessary. This is an interesting idea but not sufficiently explored in the introduction at present

Additional studies have uncovered a diverse suite of endophytic taxa on branches or tree bark, and the authors could situate their study in this literature as well. For example Lynn Boddy, Pellitier et al., etc

Line 91: it is not clear what mycorrhizae forming endophytic fungi means. Mycorrhizal fungi inhabiting an endophytic niche? The authors could consult literature by M.A. Selosse exploring the evolution of the mycorrhizal niche (i.e. the ‘waiting room hypothesis).


Methods
Please provide additional information regarding the proximity of each site to one another.
Additionally, please provide information regarding what is an ‘upper’ or ‘lower’ branch. Information on the relative age of the host trees, years since clearing of the forest, overstory composition, soil type, and other site level information is necessary. Were the bark samples surface sterilized, if not why? Were there epiphytic plants growing on the sampled tissue, or was it bare? A photograph could be added to the supplement.

This data is essential for reproducibility and context.

-The parameters used in the DADA2 command should be reported.
-Which UNITE database version was used? Please report

-Overall I am missing a section in the methods discussing the statistical tests that were employed. It appears that certain tests for alpha and beta diversity were conducted? The statistical tests need to align with the questions of interest proposed in the introduction. Please consider some of the recommendations regarding presentation of results (below) that may assist in guiding the analysis. For example, instead of an occurrence test, perhaps the relative sequence abundance of ‘orchid mycorrhiza’ across each sampling type could be provided.

Recommendations for the presentation and analysis of the results:
In the introduction, the authors stress that coffee plantations may be a ‘reservoir’ for fungal diversity. This seems plausible, however, the results do not provide context to evaluate this possibility. Comparison of overall alpha diversity after rarefaction could be provided and then compared to other studies from this region. Moreover, the relative alpha diversity of the different habitat types could be reported as a nice box plot and analyzed with ANOVA.

Consider presenting an ordination (NMDS or PCoA) visually orienting the reader to differences in beta diversity among the sample types (i.e. tree vs shrub, and upper vs lower branch). A PERMANOVA implemented using the Adonis function in vegan is suitable for such an analysis.

It would be interesting to know, given that the two sites span such a wide latitude gradient if elevation figures as a driver of overall alpha or beta diversity

Table 1 is great but can be moved to the supplement.

Table 2, 3 and 4 are useful to display the data, but i might recommend adding additional information regarding which habitat type each sample is derived (i.e. where on the bark).Especially for table 2 and 3, an indicator taxa analysis could be a useful way to assess this data.

The insights derived from Figure 1 are relatively minimal when Fungal sequences are clustered at the Phylum level. The authors could perhaps consider remaking the figure with fungal Class taxonomic assignments, which may provide more nuance to the findings. The authors say in line 206 that no other trends were observed in lower taxonomic levels? Is this a statistical result? What is driving the trend in the Figure 1 at present? Are the distinctions between Asco and Basidiomycota among the samples a function of elevation? Sampling location?

Figure 2 could be improved by collapsing the stacked bar plots into box and whiskers displaying sequences assigned to each habitat type, not necessarily presenting data for every single sample.

Experimental design

Overall the experimental design is sufficient, however few samples were analyzed, and at present the overall analysis remains relatively weak with many areas for improvement. See above.

Validity of the findings

Validity of the findings is sufficient Molecular sequencing methods are solid. Statistical method are not reported, and clear findings therefore are missing.

Reviewer 2 ·

Basic reporting

a. Table2: “Sequence counts” – is this OUT or ASV?
b. Table3: “Sequences counts”
c. Line215-220: I suggest two figures to illustrate this result.

Experimental design

Line119: I suggest describing the forward and reverse primers separately and including the primer sequence.

Validity of the findings

a. I suggest PCoA plots to discuss the beta diversity results.
b. This is a well-structured manuscript. My concern is the study used unequal numbers of samples of coffee and tree. This may rise bias.

---

## Round 0.2 · accepted · Accept

Thank you for considering all suggestions by the two reviewers, they improved very much your article. My only comment is that perhaps the legend included within Figure 2, on the right, will have to change to a larger font, but the production team of PeerJ should indicate whether this is necessary or not.

Reviewer 2 ·

Basic reporting

This is well structured and written manuscript studying the fungal community in shade-coffee plantations with ITS amplicon analysis.

References: please add DOIs to several references, e.g. Line534, Line567, Line641.

Table 1: please add the column header.

Experimental design

No comment

Validity of the findings

No comment